# Educational 5G Edge Computing: Framework and Experimental Study

Qingyong Chen [1], Zhe Wang [2,*], Yu Su [3], Luwei Fu [1] and Yuanlun Wei [3]

1   School of Computer Science and Engineering, University of Electronic Science and Technology of China, Chengdu 610000, China
2   Department of Computer Science, University of Exeter, Exeter EX4 4PY, UK
3   Education Product No. 2 Center, China Mobile (Chengdu) Industrial Research Institute, Chengdu 610000, China
*   Correspondence: zw329@exeter.ac.uk

**Abstract:** Benefiting from the large-scale commercial use of 5G, smart campuses have attracted increasing research attention in recent years and are expected to revolutionize traditional campus activities. However, there are some obstacles that hinder the practical deployment of MEC (multi-access edge computing). First, traditional information infrastructures on campus cannot support latency-sensitive and computing-intensive smart applications, such as AR/VR, live interactive lectures and digital twin experiments. In addition, the mixture of old and new applications, isolated data islands and heterogeneous equipment management introduce more challenges. Moreover, the existing MEC framework proposed by ETSI and 3GPP cannot meet the specific deployment requirements of smart campuses, e.g., educational data security, real-time interactive applications, heterogeneous connections, and others. In this paper, we propose a 5G-based architecture for smart education information infrastructure; a new dedicated cloud architecture eMEC (educational multi-access edge computing) is defined. It consists of a UGW (universal access gateway) and an eMEP (educational multi-access edge computing platform), making it possible to satisfy education-specific requirements and long-term evolution. Furthermore, we implement the framework and conduct real-world field tests for eMEC in a university campus. Based on the framework and practical field tests, we also conduct a measurement study to unveil the spatial-temporal characteristics of mobile users in the smart campus and discuss exploiting them for better network performance. The experimental results show that the system achieves satisfactory performance in terms of both throughput and latency.

**Keywords:** 5G; multi-access edge computing; educational system; eMEC; field test

## 1. Introduction

5G is expected to bring significant economic and social benefits in the coming years [1], proving to be an area of great interest that has been debated over the past few years. 5G's ultra-fast speeds, lower latency and the ability to connect a massive number of mobile devices will enable new and improved opportunities to increase equal opportunity to education with distance and remote learning. The emergence of new technologies such as artificial intelligence (AI) and big data has made a dramatic impact on the way teaching and learning activities are conducted, promoting the development of smart education and smart campuses. As a high-end form of an intelligent education system, a smart campus can comprehensively revolutionize campus activities and empower smart applications and specialized innovations in a campus. However, it is difficult for current educational communication systems based on 4G networks to meet the stringent requirements of intelligent applications in smart campuses, e.g., augmented reality (AR)/virtual reality (VR) teaching, live interactive lectures, and inspection robots, among other things. The reason for this lies in 4G's low transmission speed, poor quality and high latency, which hampers

the implementation of systems and applications in smart campuses. 5G networks are a promising way to solve the above key challenges in educational communication system. Compared with 4G networks, 5G has significantly improved the service characteristics in many aspects. Specifically, 5G supports up to 10 Gbps in enhanced mobile broadband (eMBB) scenarios, millisecond-level latency in ultra-reliable low-latency communications (uRLLC) scenarios, and 100 million connections in massive machine-type communication (mMTC) scenarios [2–4].

For higher education, 5G leverages the new platforms based on AR/VR technologies to enhance students' learning experiences. 5G can be used to live stream lessons in real-time with an UHD 8K resolution, improving access to education in rural areas. 5G can potentiate the use of robots to help students with special needs by delivering the connectivity needed for these robots to respond in real-time. The bandwidth and responsiveness delivered by 5G networks will leverage the use of AR/VR technologies in education, enhancing students' learning experience. Several use cases related to 5G in the education sector are undergoing testing or have been implemented in universities around the world, providing enhanced learning experience, such as VR/AR for education, walled-off classrooms, smart campuses, real-time location services, AI-based systems to analyse students's engagement, classroom automated attendance systems, and automation of teachers' administrative tasks [5].

Although the current 5G technology has roughly matched the baseline communication requirements in smart campuses, the rich mobile applications and demands for large concurrent services can still result in excessive computation and transmission delay, as well as excessive network traffic load. The current campus infrastructures with centralized cloud servers can hardly meet the above requirements. To support real-time and heterogeneous applications, the concept of multi-access edge computing (MEC) has been proposed. MEC enables computing tasks to be offloaded to edge servers placed at radio access network (RAN) nodes. In this way, MEC can achieve lower end-to-end latency, reduced backhaul network load, less energy consumption, and more secure data transmission.

Considering that the existing work has focused solely on high-level and abstract system design, as well as independent MEC-based resource scheduling algorithms and MEC-based smart campus applications, the structural design of 5G-MEC for smart campuses is still missing. However, this work is quite challenging and has many issues to be solved. After discussions with more than 100 higher vocational colleges and deployments in more than 30 campuses [6], the European Telecommunication Standards Institute (ETSI) has identified several key issues regarding the currently designated MEC framework [7], as well as long-term network issues, equipment management issues faced by education customers, new and old application integration problems, data island problems, and others. We list and discuss the problems that hamper the implementation and deployment of educational 5G edge computing with practical settings in the smart campus scenario.

- *New infrastructure is needed to implement a campus network that can be monitored, managed and controlled in a unified and agile manner.*
  The reason is threefold:

  (1) The numerous campus information systems can significantly degrade the available bandwidth;
  (2) Multiple access technologies coexist, including 5G, 4G, WiFi, Bluetooth, Zigbee, NB-IoT, LoRa, fiber, and more. Due to the inconsistent network management and construction methods of different campus networks, it is not easy to establish seamless communications among campus networks, which causes serious information islands and brings high operating and maintenance costs;
  (3) Furthermore, the lack of common guidelines and standards for multi-network fusion hinders the development of innovative educational applications because educational and teaching data are difficult to share among numerous educational subsystems [8,9]. Even though multiple access functions are defined in the MEC architecture [10], there is no detailed management method for different networks to deal with these issues.

- *The application authentication is not unified; thus, user data needs to be aggregated and connected* [11].

  The applications lack a unified portal, making them inconvenient for teachers to teach classes, e.g., application management is problematic. Data silos are serious, and manually reporting is inefficient in increasing teachers' workload. The lack of regional governance results in an inability to analyze data effectively. Hence, it is essential to solve the problem of a unified application portal and realize rapid deployment as well as the aggregation and display of campus data.

- *Urgent need for efficient device management to schedule computing resources.*

  Considering the resource becomes more constrained with the deployment of more smart applications, it is necessary to manage the devices and services based on the temporal and spatial characteristics of campus users, especially with the tidal effects of various campus applications [6].

- *Current MEC structures cannot guarantee data security.*

  Although many solutions have been proposed to ensure the security of educational data [12,13], the campus still intends to comprehensively supervise the 5G-based smart educational information infrastructure to fully protect sensitive data such as educational teaching and student portraits. Considering that many network elements (such as user plane function (UPF), application function (AF), network element function (NEF), MEC services, and others) are integrated with MEP and do not support independent deployment, which may interfere with the normal operation of MEP and thus render 5G networks unavailable, communication service providers (CSPs) are not allowed to conduct school-specific MEC deployments. In addition, schools prefer to conduct the necessary exchange of signaling with the 5G network rather than expose their applications, educational teaching and scientific research data to CSPs.

To address the above issues faced by practical smart campuses, this paper proposes a new system architecture—educational 5G edge computing. The proposed framework includes a dedicated network for education and a dedicated edge cloud for smart education based on MEC. For the dedicated edge cloud for education, a new MEC framework eMEC (educational multi-access edge computing) is proposed, which can provide the capabilities of an industry gateway and an educational cloud platform and can meet both campus-specific MEC deployment requirements and long-term evolution. We implement the proposed framework and conduct real-world experiments in university campuses. The results show that our system provides convenient and efficient management of the 5G edge computing systems for smart campuses. Additionally, our framework supports scenario-specific optimizations to various applications in smart campuses based on the measurement study on the temporal and spatial features of the campus users.

The main contributions of this paper include:

(1) We propose a novel infrastructural architecture for educational 5G networks, which provides a practical way to integrate cloud-edge resources.

(2) We deploy a real-world educational 5G network in the university campus and implemented the proposed architecture.

(3) We obtain spatial temporal measurement results from the deployed network with the dedicated architecture. The results show that the proposed architecture can provide important guidance to the service management, application deployment, etc. for smart campus system deployments and applications.

The rest of the paper is organized as follows. In Section 2, we introduce the related work. In Section 3, the 5G smart campus system architecture is presented, the functionality is introduced and the integration of the proposed system with the existing educational system is put forth. Section 4 presents measurement study and field test results. Finally, the conclusions introduce some other interesting findings and future works.

## 2. Related Work

Educational 5G edge computing has become a promising way to build smart campuses [14–18]. The combination of 5G and edge computing has made significant progress in the relevant work of the International Standards Organization (ISO). The ETSI introduced the framework of MEC [7]. The 5G and MEC combination scheme defined by 3GPP [19] points out that the UPF is located in the campus and close to the MEC, and the data are forwarded to the MEC edge server through the UPF's local offloading technology, UL-CL. The core network's AF function needs to be pulled down and deployed to the MEC platform. However, some technical points need to be studied in depth, such as the fact that ETSI does not introduce the position and interface of MEC in the 3GPP architecture, and there is a lack of standards in guiding practice. Many problems, including the deployment of UPF and MEC, multi-network access, and the wide-area interconnection between MEPs, are not discussed in 3GPP.

Meanwhile, several researchers have studied several key problems for educational 5G edge computing, including task offloading, 5G-MEC integration, campus activity applications, and others. Zhang et al. [20] analyzed the existing problems of 4G MEC architecture, and presented a detailed solution for the MEC platform and local offload function. However, there are no experimental data to support this new architecture and algorithm effect. Chen et al. [21] analyzed the use of 5G core network based on uplink classifier (UL CL) offloading, IPv6 multi-homing offloading or a local area network strategy to select a UPF (user plane function) local offloading solution for MEC. However, it is not a system solution for edge computing that can guide industry applications. He et al. [22] sorted out two types of six local offloading schemes from the relationship between MEC and 5G network. They analyzed the session flow of each offloading scheme and its configuration requirements for terminals and networks. Six schemes were compared to analyze the application scenarios. However, the authors do not discuss the multiple network access in MEC architecture. In order to support applications such as virtual simulations and ultra-high-definition live broadcast interactions, the work in [23] found that the 5G network still needs to solve the problem of combining edge computing and network slicing. Although the paper does not discuss the system architecture related to software and hardware, this slice control algorithm is worth learning from to optimize system performance. The study in [24] found that the combination of 5G and MEC defined by the 3rd generation partnership project (3GPP) only realizes the basic function of enabling the deployment of MEC edge nodes nearby using the sinking of the 5G network element UPF, but further research is still needed on how to integrate 5G network slicing and MEC systems fully so that they can fully utilize their respective advantages to meet the end-to-end quality of service (Qos) and guarantee security from network transmission to computing resource allocation. The paper does not cover the topics of UPF and MEC deployment practice and verification in the education industry.

Tang et al. [25] proposed a network architecture focusing on deploying an edge cloud in the medical industry. It composes a powerful cloud resource based on the application requirements of the medical industry. This architecture is instructive for the implementation of a campus-based edge cloud, while the powerful performance results in the high cost of the architecture. However, it is too expensive for customers in the campuses. In the design of the separated architecture, too much emphasis on network security leads to the weakening of flexible deployment capabilities. In addition, some of the architecture's pain points lack targeted consideration, such as education equipment management and a unified portal. These issues make the architecture detrimental to the education industry.

Considering that the existing work has focused solely on high-level and abstract system design, as well as independent MEC-based resource scheduling algorithms and MEC-based smart applications, the existing MEC framework proposed by ETSI and 3GPP and mentioned in the above studies is temporarily unable to meet the deployment's requirements of smart campus. Based on the above research, the following issues still need further research.

(1) The deployment location of UPF and MEC is unclear in current MEC structures, which cannot guarantee data security. UPF and MEC are logically separated, but there are two possible methods of deployment: combined deployment and separate deployment, which denote that the UPF and MEC are deployed in the same room or different rooms, respectively. Actually, the combined deployment method is not suitable for the education industry: if UPF and MEC are combined and deployed in the Telecom operators room, it violates the security requirements that educational data cannot leave the campus. Conversely, if the UPF and MEC are deployed in the customer's room, it is not conducive to the operation and maintenance of the operator, and will bring security risks to the entire core network. Based on the above analysis, for an educational 5G MEC framework, UPF and MEC should be deployed separately, with UPF deployed by the telecom operators, and MEC deployed in the campus's computer room. However, in a separate deployment scenario, there is currently a lack of a network and data security assurance devices and methods between the UPF and the MEC, and there is a security risk.

(2) The problem of multi-network access on campus. The architecture does not address how data accessed and transmitted by non-5G networks are handled. There are many types of terminal access technologies in the vertical education industry. In addition to 5G, there are 4G, WiFi, Bluetooth, Zigbee, NB-IoT, and wired networks. The terminal data accessed by these networks may not be transmitted to the MEP through a 5G network. In order to ensure the network and data security of the MEP, a device and method for supporting multi-network access are required, which can perform access control, traffic control, and security monitoring for terminal data of various access technology types.

(3) The problem of wide-area interconnection between different campus's MEPs. In addition to the local data offloading, the educational scenarios demand wide-area interconnection between MEPs, such as high-definition live recorded courses, virtual simulation experiments, and other scenarios between different campuses. However, the UPF defined by the 3GPP only supports the PDU session from the UE to the DN; it does not support the connection from the DN to the DN. Thus, the UPF supports the data connection only from the terminal to the MEP instead of interconnection between the MEPs. An integrated MEC's system architecture is desired to solve the above problems.

(4) The need for efficient device management to schedule computing resources. Considering that the resource becomes even more constrained if more and more smart applications are developed and deployed, it is necessary to manage the devices and services based on the temporal and spatial characteristics of campus users, especially with the tidal effects of various campus applications.

(5) The application authentication is not unified; thus, user data needs to be aggregated and connected. It is essential to solve the problem of application unified portal, single sign-on, and realize rapid deployment, as well as the aggregation and display of campus education data.

Therefore, it is imperative to explore a converged edge computing technology to meet the connection, computing, security, and other needs of education customers and to design new ICT infrastructure for education customers in the 5G era. That is the proposed architecture of eMEC (consisting of UGW and eMEP). Through the newly introduced UGW network equipment to solve the above three problems mentioned above, the last two mentioned problems are solved by eMEP.

## 3. 5G Smart Campus System Architecture

### 3.1. Top-Level System Design of Smart Campus

As illustrated in Figure 1, a new 5G-based "1 + 1 + 1 + N" overall smart education system design is proposed, which means "one educational terminal + one dedicated network + one educational dedicated cloud + N educational applications". The system

can be a new DICT (data information and communications technology) infrastructure. It consists of four layers, involving a 5G educational terminal, 5G education private network, 5G education dedicated cloud, and 5G educational applications, which will be described in detail in the following Sections 3.1.1–3.1.4. The boxes on the right side are detailed expansions of the four boxes on the left side, denoting the sample entities, applications and services. There are no communication needs between the same level boxes of the terminal, network, cloud and application layers. The communications between terminals and dedicated cloud are based on the network layer. For each instance in the educational applications, its information is handled by the following process. The terminal collects the specific application data, and the network layer transmits data. Some data are transfered to a traditional IDC server, and the rest are executed with a dedicated cloud. The cloud layer stores and calculates data, and the application layer displays data. The dedicated network has multi-access communications and thus can establish one-hop data connection with the terminals. The dedicated cloud runs various services specifically for the smart educational applications.

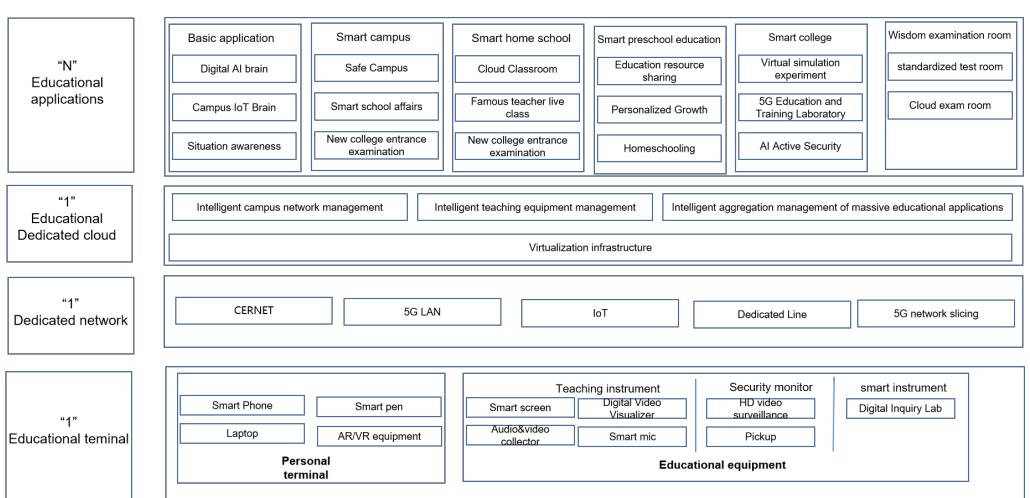

**Figure 1.** System architecture of smart educational computing.

The other three layers are orchestrated and managed by the education-specific cloud layer to enable the heterogeneous smart campus system to be manageable, flexible and applicable to different teaching scenarios and various applications inside and outside the campus. The functionality of these three layers should be exposed to educational applications through unified APIs. In this way, advanced technologies such as 5G, cloud, AI, and video in the system can be easily used by different educational applications, creating a campus-specific ecosystem of smart education capabilities for all educational service scenarios.

In addition, it contains innovative educational private network architecture and educational private cloud architecture, which will be discussed in detail in the next two chapters. The private network and private cloud can be independently or jointly used. When they are used jointly, the transmission delay can be controlled within a deterministic range to guarantee the service-level agreement (SLA) as the education data deployed on them can be directly offloaded from the education dedicated network and education dedicated cloud.

### 3.1.1. Smart Educational Terminal Layer

The terminal layer mainly uses all kinds of smart terminals to perceive the situation information of intelligent education in an all-around way to provide personalized services for teachers and students in schools. Smart terminals use 5G communication networks to achieve the interconnection and communication among people and things. In this way, the smart campuses can achieve intelligent perception, intelligent management, and intelligent services. The smart terminals are divided into personal terminals and educational

equipment. Personal terminals include smartphones, notebook computers, smart pens, VR/AR equipment, and others. Using personal terminals, teachers and students can obtain learning resources anytime and anywhere as well as interact and share learning resources with their learning peers in real-time. Personal terminal devices can also collect user data in real-time, such as cognition, behavior, learning preferences, and learning styles, among others, to help the system provide personalized learning services. Educational equipment includes teaching instruments, security monitoring, and experimental instruments with sensors. Common teaching instruments include smart microphones, smart large screens, digital video display stands, audio and video collectors, etc., which are basic tools for teachers to teach face-to-face or remotely. Security monitoring adopts high-definition video monitoring and sensor equipment, such as pickups, to collect data in real-time to provide monitoring and management around the clock to ensure campus security. Experimental instruments with sensors refer to various digital intelligent instruments, such as digital exploration laboratories, etc., which can collect and transmit experimental data to perform analysis and security control. Moreover, personal terminals and educational equipment can aggregate campus data to provide data support for smart campus management and personalized services. Although the widespread use of smart terminals on campus can provide all these advantages, a large number of them accessing the campus network results in a sharp increase in network load. Fortunately, the network access method of edge-cloud collaboration can effectively address this problem by offloading the collected 'small' data to the edge cloud close to a terminal. For example, the data collected from a smart wearable device can be directly transmitted to the near-end eMEC via Bluetooth.

### 3.1.2. Dedicated Network for Smart Education

Here, a 5G education private network is built based on the operator's ToB 5G SA core network. Combined with a multi-access edge computing base, it utilizes 5G network slicing and UPF offloading to provide a variety of network capabilities to meet the needs of educational data and public data isolation, education data transmission rate SLA guarantee, education data security, etc. It also enables the connection between multi-level clouds and multi-level education units to achieve the efficient integration of personal business and industry business. In addition, it builds mobile, secure, and reliable transmission channels for various application scenarios with low latency and high bandwidth. Furthermore, it simplifies the operation and maintenance of various educational networks by providing unified network management services via 5G, 4G, WiFi, Bluetooth, Zigbee, NB-IoT, and cable.

### 3.1.3. Dedicated Cloud for Smart Education

The 5G education edge cloud is an infrastructure and business platform specifically built for customers in the education industry. It supports the cloud-network integration, cloud-side collaboration, educational application management/service management/data management, and native security and is manageable, controllable, and perceptible to the educational network. Its key functions are three-fold: intelligent campus network management, intelligent teaching equipment management, and intelligent aggregation management of educational applications. In terms of network management, it provides unified management of educational multi-plane networks, decentralized management of educational networks by authority and domain, the platform of educational network business capabilities, and self-service management of educational network services. Through the IoT proxy technology, it can collect the usage data of all intelligent IoT devices in a region to build equipment application data indicators for network management. For the management of computing resources, it can flexibly allocate resources to education on demand. For educational application and user management, it provides an end-user-friendly application management interface, as well as educational computing and network capability opening, an educational cloud-network/cloud-edge collaborative configuration, etc. In addition, its feature of accessing all applications on campus with a unified entrance ensures

the scalable and flexible architecture of campus platform applications. It also provides a campus application supermarket for the cloud platform to realize rapid deployment, which helps school administrators to achieve unified management.

### 3.1.4. Smart Educational Application

The new 5G-based smart campus information infrastructure consisting of private networks and clouds supports the deployment and management of various applications in the 5G education cloud. Two types of applications, 5G basic applications and smart education-related applications, are shown here. The 5G basic applications are used to demonstrate the intuitive experience of 5G in improving campus productivity and transmission capabilities. The goal of education-related applications is to build a full-scenario smart campus service ecology consisting of virtual simulation experiments, high-definition live recording and broadcasting, and an AI safe campus for the whole stage of preschool education-K12-colleges and universities in various educational scenarios, such as on-campus, school-linked, and off-campus. For example, Figure 2 reveals some typical applications of smart education in campus scenarios, including VR classrooms, smart sports, cloud scoring, and webcams.

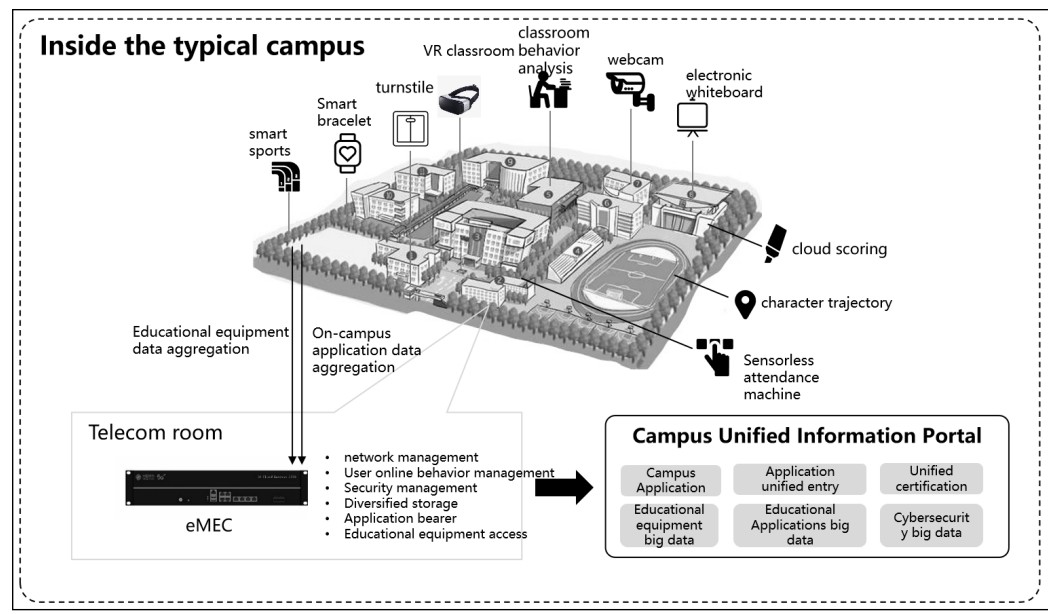

**Figure 2.** Applications of smart education in campus scenarios.

### 3.2. Smart Education Dedicated Network Architecture

Unlike a traditional education network, which is primarily focused on the network connection needs of the school, the 5G education private network can meet the basic network connection needs of the country, region, and school, as well as the requirements for data storage, analysis, and usage. It constructs a three-level business integration service system of education center cloud (central)—education regional cloud (provincial side)—5G education edge cloud (campus side). Based on the education center cloud, the government can use the basic education data of all schools to achieve evidence-based education decision-making, as well as the access services of open education services. Furthermore, the sharing of educational business capabilities, business collaboration, and the integrated development of education ecology can be realized. Based on the education regional cloud, each education management unit can learn scientific decision-making according to regional education data and regionally share basic business platforms and educational resources. Based on the education edge cloud, the 5G education private network provides the school with independently available campus resources, education data, and the control authority of the education network by realizing the basic network connection requirements of the

school. This ensures that the school can use the education network and resource scheduling on demand.

In addition, according to the different communication needs in different scenarios, including on-campus, school-linked, and off-campus, and the variability of educational terminals connected in each scenario, the local area network architecture of the 5G education network is further divided into a single-campus local education network, a multi-campus wide-area education network, a fixed-mobile integrated education metropolitan area network, and a 5G and WiFi integrated education network. For each scenario:

(1) Macro and micro base stations are used to serve as basic network access services;
(2) Cloud-network integration gateways truncate data to serve data local shunting and campus network security management and control other services;
(3) Edge computing platforms complete full business of educational data production in the school and only output business results out of school to ensure data security.

This construction method of network integration using the cloud-network integration situational awareness platform actualizes a comprehensive perception of the campus network and simplifies network operation and maintenance, thus enabling the upgrade of the campus network and the construction of a smart campus.

### 3.2.1. Local Education Dedicated Network

The local education dedicated network (LEDN) of a single campus focuses on how to satisfy the basic network connection requirements in the school to support a number of intelligent education services. The fully-connected private network for on-campus business fixed-mobile integration is an ideal solution to meet the requirements of data leaving the campus and achieve fixed-mobile integration and 5G/4G/IoT/WiFi integration. By deploying 5G macro base stations and indoor microcells on campus, the network access requirements can also be met both indoors and outdoors. When smart terminals of teachers and students connect to the wireless network through the base station for data exchange, the data are cut off and offloaded to the edge cloud platform through the industry private network gateway to alleviate the network burden. At the same time, network slicing achieves campus data isolation and SLA performance assurance, while edge computing meets the high-speed and low-latency requirements of smart educational applications. Thus, the efficient and safe upgrade of the campus information system can be realized.

### 3.2.2. Wide-Area Education Dedicated Network

In addition to ensuring network access requirements within a single campus, the multi-campus wide-area education dedicated network (WEDN) also needs to establish high-speed dedicated channels for regional interconnection, school interconnection, and mobile device interconnection to further support the interconnection of networks, services, and data across multiple campuses. By deploying a multi-level education cloud environment and private network gateway, the wide-area education private network can leverage cloud-side collaboration to achieve data distribution at different levels and wide-area interconnection of multiple campuses. Moreover, the education edge cloud can be deployed close to the main campus of each school to serve network access control while handling the school's educational business needs. For the branch campuses, their education business and data can be dynamically connected over wide areas via the 5G private network. It should be noticed that the gateway keeps sensitive data within the campus, and the rest of the data is shared across multiple campuses over a wide area network, thus ensuring the data security to a certain extent.

### 3.2.3. Fixed-Mobile Integrated Education Metropolitan Area Network

The fixed-mobile integrated education metropolitan area network takes the education cloud as the anchor point to achieve full coverage of the campus 5G wireless network and gigabit optical fiber network. By organically integrating these two networks, the communication network and information applications are highly correlated and integrated

to achieve the construction goal of cloud-network-terminal integration. The fixed-mobile integrated education metropolitan area network brings several benefits, such as ensuring the convenience and reliability of smart educational applications on campus, as well as enabling the centralized control of network performance and application content quality through the central cloud.

### 3.2.4. 5G and WiFi Converged Educational Network

The integration of 5G and WiFi is another key focus of the educational local area network. The integration of the two networks enables the full coverage of the campus network and ensures the safety, reliability, and stability of network services used for educational applications. Its core is to deploy an educational edge cloud equipped with a 5G cloud-network integration gateway near the school. In the integrated network, WiFi is used to meet the basic network requirements of specific network services, while the 5G education network serves applications with higher network requirements. For example, in general areas such as canteens and ordinary classrooms, WiFi coverage can be used to ensure normal network resource access and video access requirements; in key areas such as laboratory buildings and scientific research buildings, the 5G education private network or a 5G and WiFi hybrid can be considered. It should be noted that both types of networks can access the education edge cloud for unified IP and centralized network management and control. The 5G and WiFi converged educational network provides high security, high reliability, low latency, and a large bandwidth network service guarantee for educational applications.

### 3.2.5. Dual Domain Dedicated Network

A 5G dual-domain private network, which effectively enables college teachers and students to log in to the on-campus management system and access on-campus academic resources locally and nationally without using VPN dial-up, has recently been in demand by more and more customers. With technologies such as ULCL shunting, contracted dedicated DNN and multi-DNN shunting, campus teachers and students can access the campus intranet and the Internet without changing cards, numbers, and settings on campus, locally and nationwide. The top priority now is to build dual-domain private network services based on 5G private networks to distinguish the campus intranet and Internet access traffic for different users, as well as to mark the two types of traffic to separately meet the needs of offload billing.

### 3.3. *Dedicated Cloud Architecture for Smart Education*

As discussed in Section 2, many shortcomings of current MEC architectures prevent them from education scenarios. To solve the above problems, a novel eMEC architecture for intelligent education is proposed here, as shown in Figure 3.

This architecture falls under the 3GPP framework and also refers to the MEC definition in ETSI. In the proposed architecture, a new network element named educational multi-access edge computing (eMEC) is introduced between UPF and the data network (DN). In addition, the 3GPP network is connected with a DN (data network) through N6. The DN can be a central cloud, edge cloud and IDC (Internet Data Center). The eMEC proposed in this paper can be connected with the existing network N6 interface, and the relevant communication interface is designed. It is a new network infrastructure that integrates universal access gateways (UGW) and IaaS (infrastructure as a service)-based MEP business platforms for direct deployment of educational applications.

The edge cloud eMEC for the education industry contains four components: UGW, IaaS, the eMEP platform and educational applications on it. Among them, UGW, as a dedicated coordinator system, plays a key role in network management, wide area connection, user access control and network capability exposure. The UGW interconnects with UPF by communication interface N6 (Nx is the standard communication interface in 3GPP, defined in 3GPP ts23.501) [19]. By cooperating with eEMP, the functions of UGW

become feasible in educational applications, and, in turn, UGW further enhances eMEP so that it can be managed locally by a campus administrator. eMEC mainly provides general network and computing resources, which is also an intelligent educational application for end users. In addition, the eMEC for the education industry can be deployed in the computer room of the campus without cybersecurity risk and puts the devices under full end-user control.

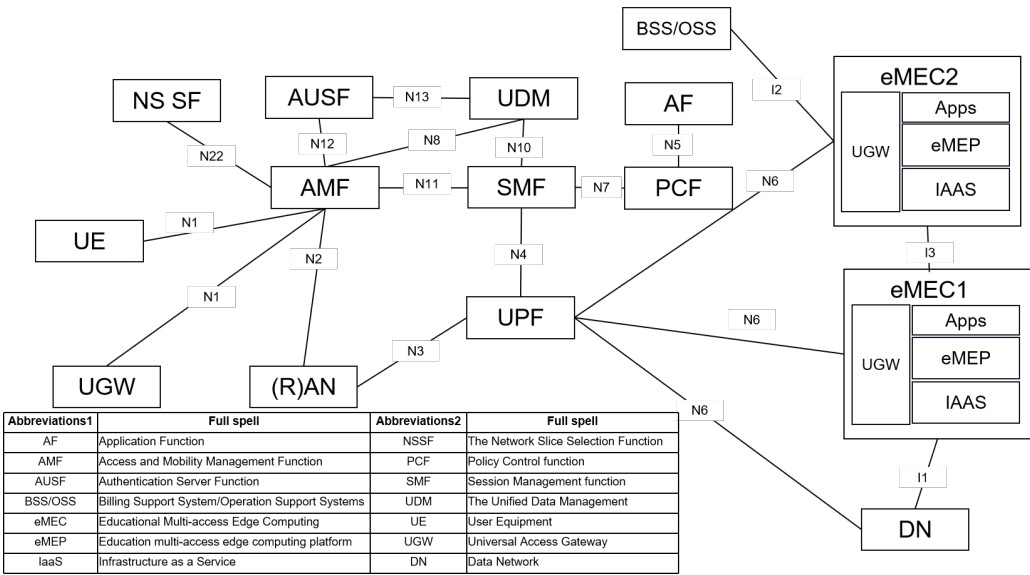

| Abbreviations1 | Full spell | Abbreviations2 | Full spell |
|---|---|---|---|
| AF | Application Function | NSSF | The Network Slice Selection Function |
| AMF | Access and Mobility Management Function | PCF | Policy Control function |
| AUSF | Authentication Server Function | SMF | Session Management function |
| BSS/OSS | Billing Support System/Operation Support Systems | UDM | The Unified Data Management |
| eMEC | Educational Multi-access Edge Computing | UE | User Equipment |
| eMEP | Education multi-access edge computing platform | UGW | Universal Access Gateway |
| IaaS | Infrastructure as a Service | DN | Data Network |

**Figure 3.** The architecture of a smart education-3GPP system.

The workflow of the proposed eMEC architecture is presented as follows: at first, terminals in the education industry are connected to the wireless network via the 5G base station (the base station is deployed on campus) to transmit collected educational data. Then, the data distribution gateway UPF receives data and offloads it to the universal access gateway UGW, which immediately routes the data to the locally deployed education edge cloud eMEP service platform.

With the help of network slicing technology, the eMEC architecture can realize the isolation of public network data and education data and meet the SLA performance requirements (rate requirement and delay requirement) of education data transmission. The introduction of edge computing technology also provides high-speed, low-latency support for smart educational applications in the eMEC architecture. The combination of the above technologies and the eMEC architecture further empowers the upgrade of the information system in schools, accelerates the rapid development of smart education and ensures the implementation of the national smart education strategy.

To summarize, the architecture shows a systematic view of the '1 + 1 + 1 + N' model, which consists of both the proposed eMEC and the existing facilities and applications. Our contribution lies in the layer of the 'educational dedicated cloud', which takes over the upstream data for local edge processing or remote cloud computing. eMEC is our solution for the educational dedicated cloud. The main contribution of eMEC is two-fold:

(1)  We propose a novel universal access gateway (UGW) providing a comprehensive software package including multi-tenant management, campus multi-standard network unified access authentication management, and wide-area interconnection management. To the best of our knowledge, there are no carrier-level gateways that simultaneously support the above functionalities due to the authority gap between the campus intranet and the backbone network.

(2)  We integrate the heterogeneous smart educational services into educational multi-access edge platform (eMEP), which is an IaaS-based MEP platform for direct deployment of educational services. By cooperating with eMEP, the functions of UGW

become feasible in educational applications, and, in turn, UGW further enhances eMEP so that it can be managed locally by a campus administrator.

### 3.3.1. Universal Access Gateway

The universal access gateway capability is a core capability on the education edge cloud. It is an industry-specific network infrastructure based on universal hardware capabilities, supporting 4G, 5G, WiFi, and other full scenarios. This infrastructure meets the operational and security requirements, and opens up local distribution of enterprise user data and network capability. It can also enable the unified access management capabilities of multi-standard networks in educational institutions, the wide-area interconnection capabilities between institutions, the choreographed decentralized access capabilities, the ubiquitous network capabilities, the service orchestration and open capabilities, the network security management, control capabilities, and the eMEC. Furthermore, it meets the collaborative needs of 5G cloud-network integration while ensuring that the core data for education do not leave the campus and are safely transmitted.

UGW functions include:

(1) Multi-tenant management: support for the configuration management of multi-tenant information, including adding, modifying, deleting, and querying functions;
(2) Access authentication: authenticating the edge cloud platform;
(3) Authorization management: supporting the service authorization management of the edge cloud platform, supporting the addition, deletion, and modification of authorization information, and enabling the edge cloud platform to actively query the authorization information;
(4) Multi-access network management: supporting unified management of multi-access networks, including binding physical ports for access networks, naming (renaming), and flow control;
(5) Shunt switch: supporting the on/off control of the shunt of the edge cloud platform;
(6) Edge cloud platform status monitoring: heartbeat detection—iMEP heartbeat monitoring, enabling edge cloud platform nodes to enable offloading when they go online and disable automatic control of offloading when they go offline; status query—supporting querying edge cloud platform online/offline status;
(7) Data security transmission: tunnel encryption configuration, supporting IPSec (Internet Protocol Security) tunnel encryption and configuration (supporting ESP protocol, realizing data source verification, data integrity verification, anti-packet replay attack, and encryption functions; supporting IKE automatic negotiation to establish tunnel; supporting DES, AES, and SM4 common encryption algorithms; supporting SHA256/384/512 and SM3 common digest algorithms);
(8) Access control management: supporting the configuration and execution of access control rules based on IP quintuple;
(9) Wide-area interconnection management: supporting the wide-area interconnection of multiple campuses, supporting functions such as static routing and NAT configuration.

To summarize, the universal access gateway (UGW) plays a key role in network management, providing nine main functionalities: multi-tenant management, campus multi-standard network unified access authentication management, wide-area interconnection management, authorization management, a multi-access network, a shunt switch, edge cloud platform status monitoring, data security transmission, and access control management. Among them, the first three modules are uniquely proposed and not supported by the existing works. The reason is that it requires the interplay between the authentications of the campus network and the carrier backbone network. By bridging this authority gap, we can support the three modules simultaneously in UGW.

### 3.3.2. Education Multi-Access Edge Computing Platform

The educational multi-access edge computing platform (eMEP) is divided into five layers from the technical structure, including the unified platform management layer, the

platform capability open layer, the platform capability access layer, the basic technology service layer, and the infrastructure layer. These five layers work in concert to provide the following functionalities:

(1) Providing IaaS virtualization services where edge computing IaaS serves edge applications in the form of cloud and providing a basic operating environment for hosted applications;

(2) Providing edge computing platform services, including supporting edge application deployment, providing a unified entry for application deployment; supporting configuration and distribution of offloading equipment and reporting offloading configuration-related information to the private network operation management platform; and enabling the ability of the wireless-side or core network-side interfaces to provide services for industrial applications, including location services, bandwidth management services, and wireless network information. Based on specific business scenarios, it can also provide capabilities such as video streaming and AI algorithm libraries for applications to meet the corresponding application requirements;

(3) Providing Internet business access services, including providing Internet application access lines and ensuring access security; supporting Internet application deployment and providing a unified portal for Internet application deployment; and supporting Internet application orchestration and management;

(4) Providing central cloud access capabilities and services, including cloud-edge collaboration services; supporting edge cloud to leverage the powerful computing and storage capabilities of the central cloud; providing rapid deployment of applications to edge cloud services through central cloud; providing rapid deployment of AI algorithms to edge cloud services; and providing rapid deployment of big data models trained on the central cloud to edge cloud services.

To summarize, educational multi-access edge platform (eMEP) is an IaaS-based MEP platform for the direct deployment of educational services. By cooperating with eMEP, the functions of UGW become feasible in educational applications, and, in turn, UGW further enhances eMEP so that it can be managed locally by a campus administrator.

### 3.4. Education Dedicated Cloud Integrating with Network and Existed Educational System

3.4.1. Integration with Education Dedicated Network

The eMEC system cannot connect to other networks devices or elements without UGW, except those interfaces that were defined in ETSI and 3GPP. The additional I1–I3 network interfaces are introduced here, and UGW can also communicate with AMF through an N1 network interface, as shown in Figure 3.

The I1 is defined as the interface between UGW and third-party data networks (DN), which is mainly used for access authentication, network control, and data transmission. For example, the UGW can recognize and allow application access to the third-party data networks by active or passive access authentication; the third-party network's bandwidth can be controlled by UGW for flow control and load balance. Additionally, the I1 interface carries the data transmission between UGW and third-party networks through dedicated tunnels, and these tunnels realize the secondary encapsulation of network data to ensure data security, unified protocol parsing, and other essential functions. The I2 interface is the management interface for UGW's policy configuration and operation status monitoring, including the UGW registration, operation authority, method and range for network capabilities exposure, status monitoring, etc. All UGW's related interfaces are managed by the I2 interface, including I1 and I3. The I3 interface realizes wide-area interconnection between UGWs when differents eMECs deploy in different campuses. It can be implemented based on traditional SD-WAN and can be 5G LAN or other technologies, allowing interconnection between campus at anytime and anywhere, without the conventional line's limitation and affordance.

### 3.4.2. Integration with Existing Educational Systems

The introduced platform, eMEP, which deploys on IaaS, can load many educational systems and apps. It will work with UGW as a system and integrate with existing educational systems or systems in different campus. Many educational sub-systems that exist in campuses do not support cloud-specific deployment and data sharing between applications; it is thus difficult to move those existing systems or educational applications to a 5G-based education dedicated cloud. A new strategy for the integration of the 5G system and the existing educational system is needed to promote innovative educational applications from demonstration to practice.

On the one hand, for the new infrastructure, it is recommended to implement an integrated strategy that matches customer needs based on the eMEC architecture (dedicated deployment for educational customers), and then deploy the appropriate 5G educational and teaching applications (such as virtual simulation platforms, AI safe campuses) in eMEC according to their exact requirements. On the other hand, traditional education and teaching applications can also remain on the original IDC infrastructure without migration and be centrally managed through the eMEC portal. In this way, as more and more educational applications become 5G and cloud-based applications, the traditional campus information infrastructure will naturally be eliminated and integrated into the 5G information infrastructure. In addition, to efficiently support different campus scenarios, the edge cloud eMEC in the education industry should be deployed in schools. If the service capability of a single eMEC is assessed to be insufficient, multiple eMECs can be stacked to provide more powerful computing power.

## 4. Measurements and Experiments

Extensive measurements and experiments were conducted in a real-world campus to highlight the necessity and advantage of our educational 5G edge computing. The traffic data from five base stations in this campus, which are located in the vicinity of a hospital, classroom, gym, laboratory, and lakeside park, were analyzed. The per-hour data were collected on both weekdays and weekends. On the one hand, the measurement study in Section 4.1 reveals the correlation features of campus service. On the other hand, experiments in three scenarios are implemented in Section 4.2 to evaluate the overall performance of the proposed system. The specific experiment setting and analysis of the results are detailed in the following subsections.

### 4.1. Measurement Study of Campus Service

Figure 2 refers to a hypothetical campus as a case study. It shows that in a campus, different buildings and areas are given specific and distinguished functionalities with different requirements regarding communications and computing. For example, in a VR classroom, the real-time computing requirement will be intensive. In a gym, only delay-tolerant trajectory services are required. This characteristic should be supported and captured by the educational network, which further supports application-level service and resource management.

General research shows that users on campus have strong temporal and spatial regularity, but this has not been quantitatively analyzed in research on smart educational applications. The research team conducted long-term data research on a well-known university in western China (with a population of no less than 30,000 people on a single campus). The data dimensions included, but were not limited to, the spatio-temporal activities of personnel, the curriculum arrangement of the university, the distribution of operator infrastructure in the university, operator data statistics, the spatio-temporal characteristics of campus users' traffic, the characteristics of campus users' uplink and downlink data, etc. [26] The heatmaps of the number of connected users are depicted in Figure 4, which reflects the temporal and spatial differences. In addition, the quantitative change of throughput and user connections in 24 h are shown in Figures 5–7. Some interesting characteristics of network servers in campus can be observed:

- *Spatial correlation:* the distribution of service is correlated with geographic location, where most of the connections are located in functional areas, such as the classroom, lab, and library.
- *Temporal correlation:* the network activity is correlated with the school timetable, where the network connection and servers are concentrated at duty time on weekdays and afternoons on the weekend.
- *Combined correlation:* The distribution of network services is affected by the combined correlation of both spatial and temporal elements. For instance, the peak of network traffic is found in the classroom or laboratory at duty time on weekdays and in the lakeside park on the weekend afternoon.

In summary, there is indeed a significant spatio-temporal correlation in network services for mobile users in campus, which provides heuristic information for a series of network optimizations, such as network configuration, network slicing, resource scheduling, and channel allocation, in a flexible and efficient manner.

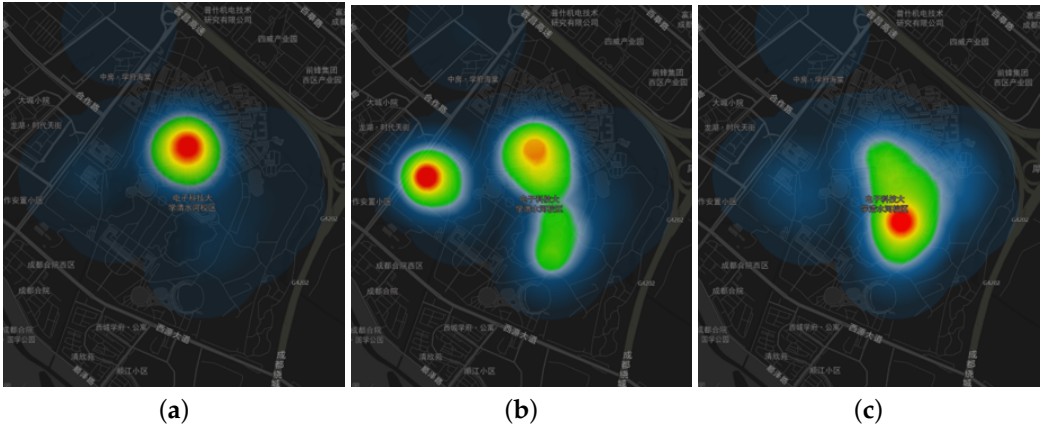

**Figure 4.** The heatmap of connected users in real-world campus scenarios: (**a**) 10:00 am on a weekday; (**b**) 1:00 pm on a weekday; (**c**) 5:00 pm on a weekend.

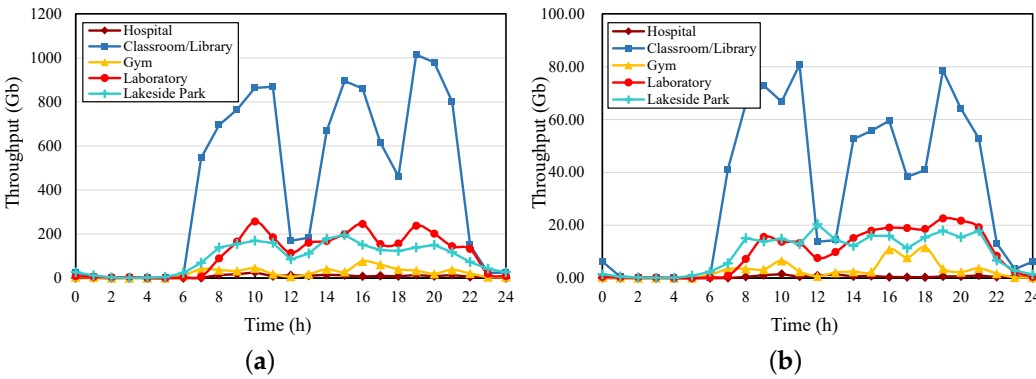

**Figure 5.** The total throughput of diverse base stations within one weekday (24 h). (**a**) Downlink; (**b**) uplink.

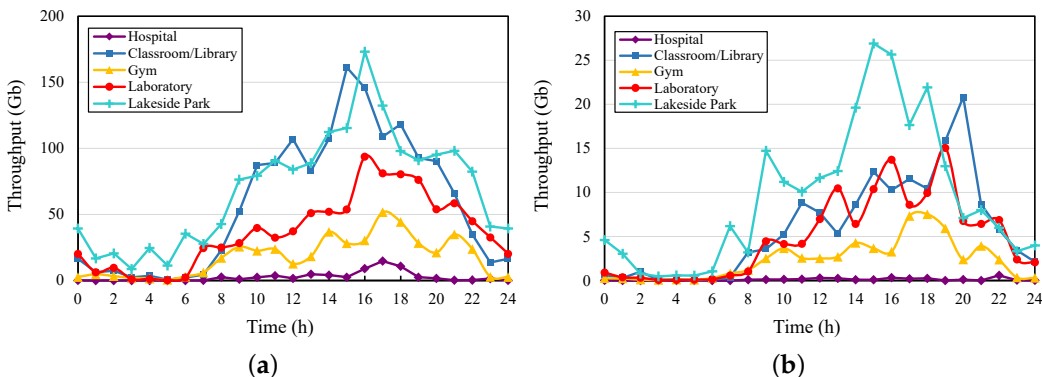

**Figure 6.** The total throughput of diverse base stations within one weekend (24 h). (**a**) Downlink; (**b**) uplink.

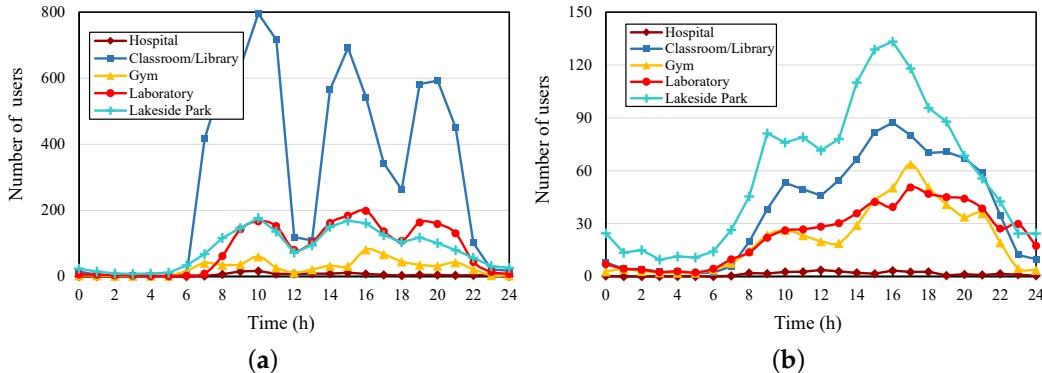

**Figure 7.** The average number of connected users (24 h). (**a**) Weekday; (**b**) weekend.

### *4.2. Evaluation of our Proposal*

#### 4.2.1. Education Features Overview and Comparison

The system architecture proposed in this paper addresses the ETSI MEC problem in the smart education scenario introduced in Section 1, where a network slicing-based education private network and an edge cloud are designed to jointly support different campus services. Benefiting from the introduction of the eMEC, the system can guide the school to complete the deployment. The traditional architecture only supports public 5G but not MEC, and campus applications are only deployed in private clouds. To demonstrate the advantages of the proposed system, we listed 13 functionalities supported by the proposed architecture; the comparison with the traditional architecture is shown in below.

- Fully support 2 functionalities: authorization management, multi-access network;
- Partly support 6 functionalities: shut switch, MEC platform status monitoring, data security transmission, access control management, IaaS virtualization services, MEC platform services;
- Does not support 5 functionalities: multi-tenant management, campus multi-standard network unified access authentication management, wide area interconnection management, Internet business access services, and central cloud access capabilities and services.

#### 4.2.2. Field Test Scenario Overall Designing

To verify the availability of the 5G smart education architecture and test its basic performance, we cooperated with a well-known university to build a demonstration project and carried out several field tests based on the project.

The 5G campus private network, cloud, and typical college education applications were deployed, similar to an imaginary campus as shown in Figure 2. In addition, another

vocational education college also participated in the deployment to test the virtual simulation experiment platform service. There were two types of clouds designed and built. One was the education edge cloud, eMEC, consisting of UGW and eMEP, which was deployed in the university campus; the other was the education center cloud consisting of multiple sets of eMEC, which was deployed in the western cloud computing center for coordinating educational services among multiple schools and for exchanging educational and teaching data. The project also deployed dedicated educational networks including LEDN and WEDN to serve in-school, inter-school, and out-of-school education and teaching scenarios. Field test instructions of the system architecture are shown in Table 1.

All education and teaching equipment in the project used dedicated access points to access the educational private network through 5G CPE, 5G modules, or other access technologies. At this stage, only basic key performances were tested, including upload data rate, download data rate, delay, and jitter. The corresponding requirements for these key performances can be found in definitions by China Mobile. In this case, campus-specific and education-specific clouds, LEDN, WEDN, public and personal educational end devices, and educational applications such as virtual simulation platforms were all involved in on-campus field testing.

**Table 1.** Field test of system architecture.

| Test Description Items | Traditional System | Proposed System |
|---|---|---|
| Access technologies | 5G, WiFi | 5G, WiFi |
| Wide area connection | Fiber optic | 5G, fiber optic |
| Communication resource | Sharing public network | Dedicated network |
| Computation resource | Physical resource | eMEC |
| Educational application | Virtual simulation test/VR for education/4K live broadcast | |
| | Estimated rate/latency:<br>Virtual simulation test (Frame rate: 50–60): 20 Mbps for 2K, 30–40 Mbps for 4K, 80–100 Mbps for 8K, $\approx$10 Mb/s for HD H.265/HEVC; and 100 ms end-to-end latency is required | |
| | VR for education: 30 Mbps (bandwidth/user), 40 ms end-to-end latency | |
| Test use cases | Intra-campus, inter-campus, out-campus | |
| Performance metrics | Upload and download data rate, latency | |
| Test time | 10:00 and 13:00 on weekdays and 17:00 on weekends | |
| Test place | Information center of a university | |
| | Simulation laboratory of a university | |
| | An office outside the university | |
| Network access point | 10 m near the 5G sub-base station's antenna | |
| Test devices | Tablet PC, PC, VR glasses, 5G CPE (5G SA sim card) | |

In particular, we tested the system performance of handheld educational devices in different aspects, taking into account various scenarios such as fixed and mobile scenarios and the presence of traffic congestion, where the data rate and end-to-end latency results are shown in Figures 8–10.

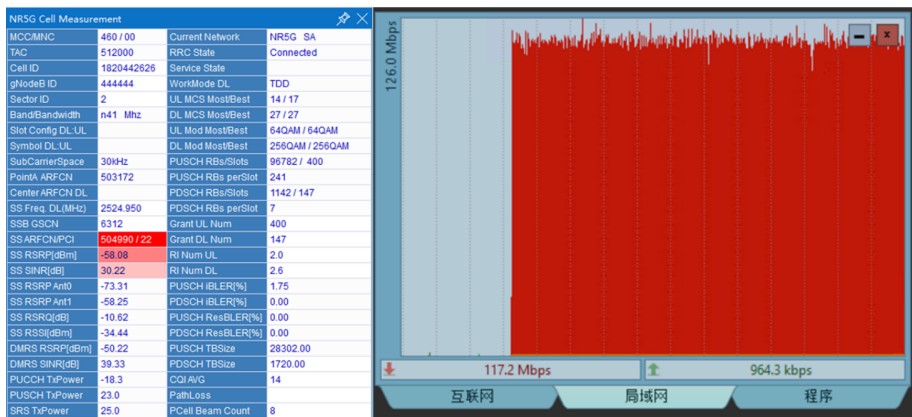

**Figure 8.** The system upload data rate performance.

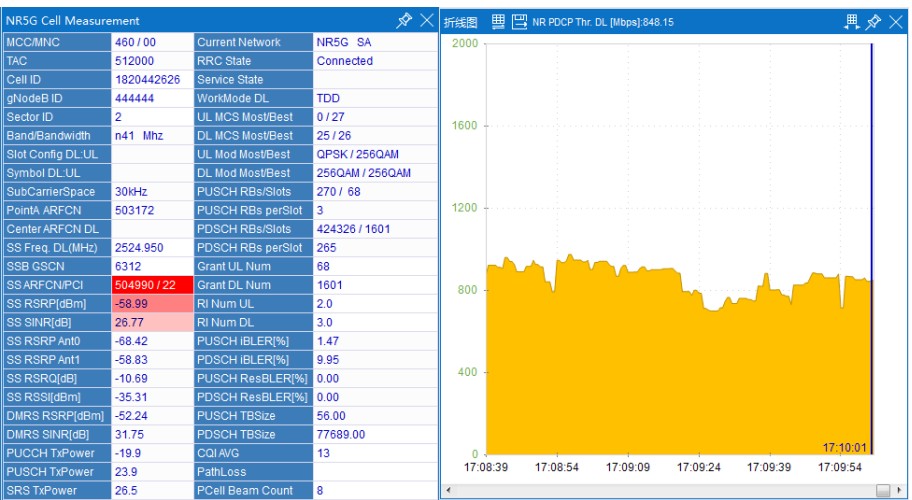

**Figure 9.** The system download data rate performance.

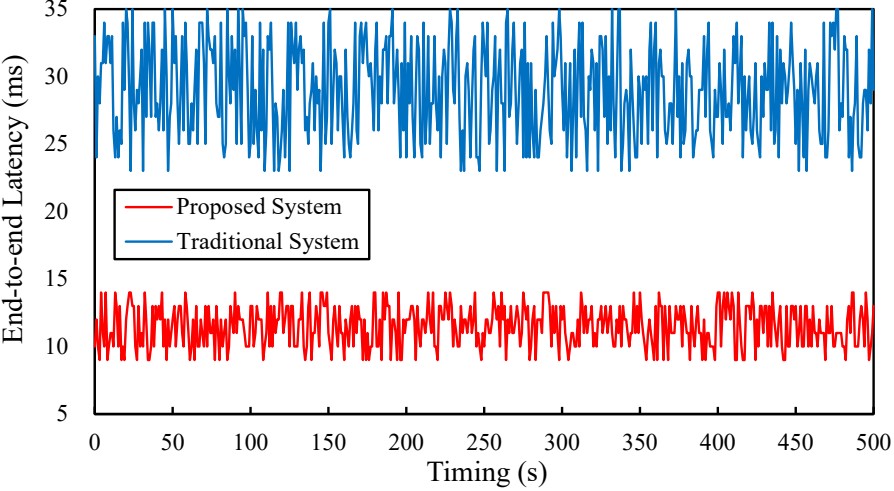

**Figure 10.** The system end-to-end latency performance.

### 4.2.3. The Proposed System Data Rate Testing

To verify the performance of the proposed system architecture, a series of tests were carried out according to the following methods:

(1)  We prepared test devices (5G UE with 5G SA sim card). Test locations were the information center of a university, simulation laboratory of a university, and an

office outside the university. The network access point was 10 m near the 5G sub-base station's antenna; the test time was 10:00 and 13:00 on weekdays and 17:00 on weekends. We tested 9 rounds at 10:00, 13:00 and 17:00 in 3 different places;

(2) We performed the FTP download service to the 5G UE on eMEC;
(3) After the rate became stable, we recorded the average rate, maximum rate, minimum rate, and air interface channel quality, e.g., reference signal receiving power (RSRP), received signal strength indicator (RSSI), signal-to-interference plus poise ratio (SINR) within 1 min;
(4) We stopped the downloading;
(5) We performed an FTP upload service to the 5G UE on eMEC;
(6) After the rate is stable, we recorded the average rate, maximum rate, minimum rate, and air interface channel quality within 1 min;
(7) We stopped uploading;
(8) We exited the test;
(9) We adjusted the test point position from the base station and repeated the above test 8 times.

In the proposed system data rate testing, there is a certain rate difference in the test results at three places. Meanwhile, the change of time has little impact on the test results. From the analysis, 5G private network users have not reached the design capacity, rate difference caused by 5G network environment at different test points. Therefore, representative results are shown in Figures 8 and 9. It is close to the mean value. This test value was obtained at 17:00 pm in the information center of university.

The results show that our architecture provides better coverage of the 5G campus private network and the channel quality of the air interface (SINR = 31.75; RSRP = −58.83 dBm). In the architecture, ordinary test users can obtain a stable 126 Mbps rate in the uplink and a higher stable rate, 800 Mbps, in the downlink. It is not difficult to find that the architecture can meet the uplink and downlink rate requirements of various educational applications defined by the standard [27].

### 4.2.4. The System End-to-End Latency Performance Testing

Some relevant test description item test devices are shown in Table 1. We tested 9 rounds at 10:00 am, 13:00 pm and 17:00 pm in 3 different places. The test values' differences are small for the three different places, and the change in time has little impact on the test results. The analysis shows that two systems have not reached the bottleneck of network capacity and performance. Here, the average value of multiple rounds of tests is shown in Figure 10.

Experimental results of the user end-to-end delay tracking running on the virtual simulation platform show that the end-to-end delay of proposed system is about 10 ms. Compared to the 30 ms delay of the public network, our system significantly reduces the network latency. This is because our system architecture has a near-end shunt, which avoids data returning to the core network for processing.

Moreover, from the field trail results, the proposed system can stably provide an ≈800 Mbps downlink data rate, a ≈126 Mbps uplink data rate, and ≈10 ms E2E latency, which can support almost all innovative smart educational applications defined by ChinaMobile in [27].

### 4.2.5. The Virtual Simulation Test on Different Systems Performance Testing

By comparing the virtual simulation experiment's effect on colleges under different architectures, we can test the differences in network, computing, and other capabilities. To test the system performance under different concurrent numbers, we conducted virtual simulation under different terminals (PC, VR glasses, 5G CPE(5G SA sim card)) at three places. The time was 10:00 a.m. on a working day, and the statistical result is the average value of 20 independent runs. The test simulated different numbers (5 users to 100 users, concurrently) of students carrying out virtual teaching in various scenarios under the

normal teaching mode. The end-to-end delay of the virtual simulation application is within 3 s, and the application runs smoothly. See Figure 11 for test results.

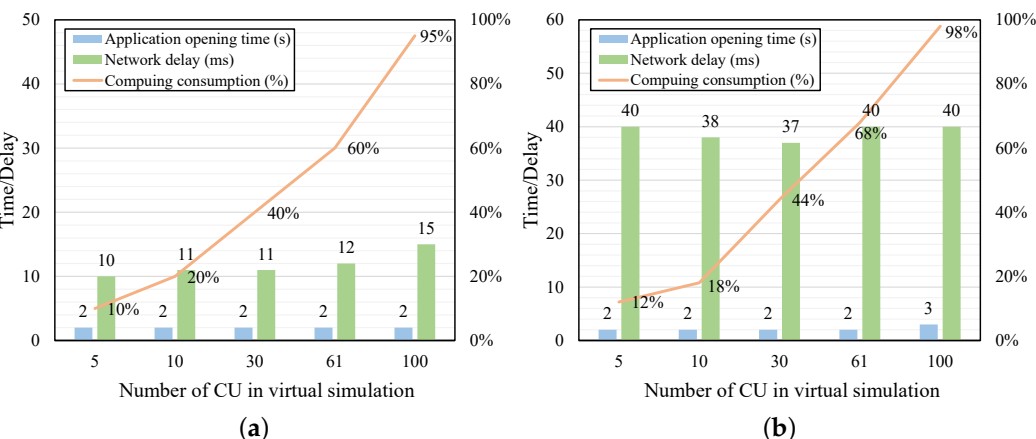

**Figure 11.** Performance of virtual simulation. (**a**) Our proposal; (**b**) traditional system.

Under our proposal architecture, the access delay is about 10 ms, the application runs smoothly, and the user does not feel any delay. As the number of concurrent users increased from 5 to 100, the application open time did not change obviously, eventually rising from 2 s to 3 s. The network latency increased slightly, from 10 ms to 15 ms, and the computing power consumption increased from 10 percent to 95 percent. Application opening, delay test, computing power consumption, etc. are shown on the left side of Figure 11.

Under the traditional architecture, the access delay is about 40 ms, and the application runs smoothly without affecting the users at all. As the number of concurrent users increased from 5 to 100, the application open time did not change obviously, eventually rising from 2 s to 3 s. The network latency does not change obviously, fluctuating around 40 ms, and the computing power consumption increases from 10 percent to 98 percent. Application opening, delay test, and computing power consumption are shown on the right side of Figure 11.

Based on the above analysis, we can conclude that the computing and storage capacity of different architectures mainly comes from the gap in hardware resources. Under similar configurations, this part has the same ability to provide applications. The end-to-end network delay in the proposed system is much better than the traditional systems. The gap mainly comes from the impact of data offloading on the network delay brought by the design.

In the current situation that colleges do not comprehensively upgrade the existing system and hardware, the compromise business optimizations include:

(1) Viewing the control delay through the end-to-end delay of the network;
(2) Modifying the network bandwidth/video frame rate of different clients to improve the fluency of course control;
(3) Optimizing the configuration encoding process and reducing the amount of transmitted data.

These operations improve the processing capacity of the server (that is, the concurrent performance of the system) and reduce the delay at the same time.

## 5. Conclusions

In this paper, we propose a 5G-based architecture for smart education information infrastructure; a new dedicated cloud architecture, eMEC, is defined. It consists of UGW and eMEP. As a novel solution, eMEC is tailored for the smart educational network, which bridges the authority gap between the campus network and the backbone network, thus covering all necessary management functions required by educational networks and

supporting heterogeneous multi-access devices in smart campus. We deploy a real-world educational 5G network in a university campus and incorporate eMEC into the network, which can effectively provide management and computing services to around 20,000 users in the campus.

Based on the experimental results from extensive field tests, we observe that:

(1) The real-world field tests show that the proposed architecture is practical and can successfully integrate various terminals and edge-cloud resources;

(2) The architecture supports both large-scale and small-scale applications by routing the user data/requests to edge and cloud according to its application-level requirements. The reason is that the multi-access dedicated network is able to directly collect and process the data and requests from the front-end terminals;

(3) We implement the framework and conduct extensive field tests, which reveals the unique spatial and temporal characteristics in the smart campus.

These observations allow us to design appropriate resource and service management schemes in our future work.

**Author Contributions:** Methodology, Q.C.; Validation, L.F. and Y.W.; Formal Analysis, L.F.; Investigation, Q.C.; Data Curation Z.W.; Writing—original Draft Preparation, Y.W.; Writing—review and Editing, Z.W.; Supervision, Y.S.; Project Administration, Q.C and Y.S. All authors have read and agreed to the published version of the manuscript.

**Funding:** This work was supported by the National Key Research and Development Program of China (No.2020YFE0200500) and the Natural Science Foundation of Sichuan Province (No. 2022NS-FSC0885).

**Data Availability Statement:** Not applicable.

**Conflicts of Interest:** The authors declare no conflict of interest.

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
