# Peer review of "Educational 5G Edge Computing: Framework and Experimental Study"

_electronics, doi:10.3390/electronics11172727_

Round 1

Reviewer 1 Report

This paper presents an architecture for smart campuses gathering educational frameworks based on MEC. The paper's readability and presentation are ok despite some typos (e.g., delicated, transmiting, stablitlity). I recommend proofreading written English. Sometimes I felt using the singular word campus could sound better in the plural (campuses), sometimes the opposite. The words "hinder" and "realize" are overused during the text. I suggest adopting synonymous. The first phrase of Section 2 is a hard-to-read sentence.

The paper presents many acronyms that could not be usual to some readers (some of them are usual, but for some readers may not). So, it is essential to write their meaning and definitions as well. The acronyms found are: eMBB, mMTC, uRLLC, UPFs, RAN, 3GPP, QoS, ETSI, DICT, SLA, AI, ULGC, DN, IAAS, eMEP, LEDT, WEDT, DED and so on. Figure 3 shows a lot of acronyms interconnected by other acronyms: N1, N2, N3, N4, N5, N6, N7, N8, N10, N11, N12, N13, N22, and I2. It is not clear their meaning and interconnections considering the whole approach.   

Besides, Figures 2 and 4 are never referenced in the text. The same occurs to other figures. Figures 2 and 3 in some parts of the text seem to reference Figures 8 and 9. Table 1 is referenced but never appeared in the text.

Figure 1 shows the proposed system architecture. How do the boxes communicate with each other? The boxes Education Terminal, Dedicated network and clouds, and education applications are explained in the text. But the other boxes don't. For example, It is not possible to see functions such as Intelligent Campus network Management, Intelligent Teaching Equipment Management, and Intelligent Agreggation Management of Massive Educational Application in the architecture. What is a 1+1+N  architecture? What is native A secure and a one-stop new DICT?

Yet concerning the architecture, the authors assert, "It also opens up the connection between multi-level clouds and multi-level education units to realize the efficient integration of personal business and industry business, and build mobile, secure, and reliable transmission channels for various application scenarios with low latency, high-bandwidth network requirements.". Then, how did the authors deal with these issues?

Besides, the authors say, "the network access method of edge-cloud collaboration can effectively solve this problem by transmitting the collected "small" data to the near-end edge cloud in real-time.". But how can it be done? It is not clear as well.

In fact, the four main boxes of the architecture are too generalistic. They do not provide enough technical details on implementing and solving the four problems-topics in the Introduction. One question is: how the architecture is going to work? What is the flow of information (data) between layers? It all should be emphasized.

Technologically speaking, is there any middleware or service running? Did the authors develop an algorithm, middleware, service, or any other technological product to allow the architecture to work? What are the paper's contributions?

Connecting existing technologies to provide a solution is an interesting approach commonly accepted in many journals. However, research papers must have "their experimental and theoretical results in as much detail as possible.". Despite the results of the experiments section, more details are lacking in the architecture implementation. It is missing a comparison with some related works and architectures that try to integrate technologies to provide services in smart campuses or related domains.

The conclusions section should be improved. The authors must insert from 3 to 5 findings of their work and some interesting discussions and future works.

Author Response

We are very grateful to the reviewer for your thorough review and constructive comments. We have carefully addressed the review comments and improved the quality of this paper. Please find below our response to review comments point-by-point.

Reviewer 2 Report

Dear authors, 

We find the topic of this research interesting and relevant. Nevertheless, we consider the following improvement opportunities:

 (1) No related work was provided. We encourage the authors to include a section of related work introducing 5G technologies, use cases where gaps are specified that would require its use, and use cases where it was applied to solve similar issues.

 (2) In the abstract, the authors mention that the "5G MEC framework proposed by ETSI and 3GPP cannot meet the specific deployment requirements of smart campuses", but we did not find a detailed description of the smart campuses' requirements, and of the 5G MEC framework. We encourage the authors to devote a section to this analysis, providing insight on constraints and how the proposed architecture addresses them.

 (3) The authors validate the whole architecture with results concerning data upload/download performance. While such a quantitative evaluation is important, we consider that more context must be provided to the users. It would be useful to (a) have a schema of the campus, where are the connection spots located and where did the authors perform the measurements. In addition, we consider it would be important to specify (a) if the measurements were performed in a single place or in multiple places, (b) how did performance change, (c) if any hypotheses for the performance deviations exist.

 (4) Regarding the experiments, it would also be relevant to inform when were they performed. Furthermore, it would be relevant to realize them at different moments of the day and week (e.g., weekday vs. weekend, and peak hours vs. non-peak hours). Furthermore, we expect the results between these times to be contrasted for further insights and findings.

 (5) In the abstract, the authors mention that they "incorporate edge computing to the existing infrastructure for connection, computing and security". Nevertheless, the paper's results validate only the connection aspect, and no details on how the existing infrastructure is enhanced in the computing and security aspects. We would encourage the authors to (a) provide greater detail regarding these two aspects and provide some means to validate them, or (b) more clearly state the limitations of the existing paper regarding these two characteristics.

 (6) Figure 1 has no relation to the descriptions provided in the Section 2 subsections. Ensure the components described are correctly mentioned and described in the text.

 (7) Line 140 mentions that "A new '1+1+N' smart education cloud-network integration service capability system" was built. Nevertheless, in Figure 1, we find three "1" layers and one N layer. Should this be named "1+1+1+N" instead of "1+1+N"? Why not?

 (8) Figure 2: do the legends in different colors have a specific meaning? If yes, please provide a reference to correctly interpret them. If not, please provide the legends in black color.

 (9) Figure 2 refers to a hypothetical campus of the campus described in the use case study? Provide a more detailed description to understand the purpose of the Figure.

 (10) Figure 3: What do the labels Nx stand for? Why do some of them repeat and some do not? 

 (11) Through the text, we found several inaccuracies/typing errors/etc. We ask the authors to correct them as listed below:

  (11a) Line 102: workloads; The lack -> "workloads; the lack" or "workloads. The lack"

  (11b) Line 116:  Considering that many network elements (such as UPF, AF, NEF, MEC services, etc.) -> replace "etc" for "and others" or similar. Introduce acronyms.

  (11c) Line 122: their applications and educational teaching and scientific research -> their applications, educational teaching, and scientific research 

  (11d) Line 143: DICT infrastructure -> introduce the acronym

  (11e) Line 151: (eg, Restful, JSON, etc.). -> JSON is not an API, but a data-interchange format. Replace "etc" with concrete examples or "among others".

  (11f) Line 256: For each scenario, Macro and micro -> For each scenario, macro, and micro

  (11g) Line 329:  Generally, dual-domain private network service requirements can be divided into three categories. -> This sentence is disconnected from the context. Please detail the three categories and how do relate to the rest of the paragraph, or remove the sentence.

  (11h) Line 351: as follows: At first -> as follows: at first

  (11i) Line 499: (CSI-RS\SSB\SINR) within 1 minute. etc. -> what does the "etc" refer to? Please replace with concrete thought.

  (11j) Line 504-506: It is not difficult to find that the architecture can meet the uplink and downlink rate requirements of various educational applications defined by the standard. -> What are the specified constraints? Please provide them beforehand so that we can contrast the results.

  (11k) Line 516: educaion devices -> education devices

  (11l) Line 518: aggregation, secondary gNB etc., -> replace etc with concrete example or use "and others."

 (12) In addition, we would ask the authors to (a) review all the acronyms making sure they have been properly introduced thorough the text, and (b) avoid the excessive use of "etc" (eventually avoid it), replacing it with more concrete concepts or replacing it with "and others", "among others", or other suitable expressions.

Author Response

(The authors gave the same response as above.)

Reviewer 3 Report

Campus networks are heavily researched areas, even in the field of 5G. There are numerous challenges regarding this topic, regarding traditional "consumer" networks as well as industrial campus networks. The concept of an educational network is interesting since it inherits characteristics from both well-known network types. Due to this, the current study is relevant and matches the journal's scope.

The paper proposes a novel 5G system architecture for smart campuses that solves a certain pre-defined problem set, focusing mainly on the MEC concept and introducing a new network element eMEC. The proposed solution also includes concepts aiming to meet different requirements that can be specific for a university campus network but also for industrial purposes.

The paper is well-written, it easy to read, and the structure is logical. The introduction (which includes related works) part is complete and not only cites relevant studies but also synthesizes them to provide a concise yet detailed set of challenges regarding the current topic.

The main part which presets the actual architecture is very detailed, and it is easy to get the full picture. However, the design methodology and the scientific soundness are missing. I consider this topic very important that can help the scientific literature to grow, but in itself, it misses the "whys" and design process that shapes the presented solution.

The evaluation part supports the conclusion, but it is a bit blurry. It would be much better if the results were compared to other systems' performance of the requirements - which are mentioned, but the numbers are not presented. Also, it would be nice if figure [8, 9, 10] could share a unified style such as the previous figures that have excellent quality. Moreover, in the end, it is hard to follow which features/devices fulfill which requirements, so I recommend summarizing them in a table.

To conclude, I don't find the paper academic, but it is a fine contribution to scientific literature, and also it is a quality paper that requires some extensions. Overall, I recommend accepting it after the authors address the concerns above.

Author Response

(The authors gave the same response as above.)

Round 2

Reviewer 1 Report

The paper was significantly improved in this version. But some issues still need to be clarified. In Section 3, the authors explain several elements of the architecture with many technologies, but none are technically explained. So, the authors should emphasize their contributions in each layer. For example, the 5G Education Edge Cloud is an authors' contribution? Did the authors create the Universal Access Gateway?

If Figure 2 is a hypothetical scenario, it should be used to cleverly guide all the explanations of technologies and layers with examples. Or, it could be moved to Section 4 or 5. In the Conclusion section, the authors affirm that eMEC has consisted of UGW and eMEP as main contributions. These last two components should be detailed since they were briefly explained in section 3.3.

Some minor issues the authors should be aware of: Several cases of acronyms in the format AI (Artificial Intelligence) should be contrary, for example, Artificial Intelligence (AI): l.25, l.31, l.37, l.38, l.66, etc. Other acronyms without information exist: UPF, AF, NEF (l.103), etc. The phrase in line 89 should be improved. The authors emphasize too much some affirmatives using words such as significant, great, massive, much lower, extremely inconvenient, urgent, relevant, deeply, and so on. The paragraph beginning in line 148 is too way big. It could be split in two. The authors should prioritize using "paper" instead of "article". Spaces are absent between some words: l.31, l.58, l.66, l.248, l.270, etc. Table 1 has no ordination, but it is unnecessary once all abbreviations are explained in the text. Is it sink or synch in line 143?

Author Response

Thank you again for your thorough review and constructive comments. We have carefully addressed the review comments and improved the quality of this paper. Please find below our response to review comments point-by-point.

Reviewer 2 Report

Dear authors, 

We thank you for providing an improved version of the manuscript. We have revised it and found some additional opportunities for improvement:

Important notes:

(1) We consider Comment 4 from the first review was not properly addressed. While additional details were included, the results were not presented, so the correlation between experiment settings and results would be obvious to the reader. Please provide additional details so the reader can better understand, e.g., if measurements differed across locations, whether there were significant differences among them, and if the reported values are an average of the measurement values obtained across the three places. Do the three places behave the same over time (e.g., peak time values)?

(2) Please improve the conclusions: some statements may be very well known to the readers, e.g., "The test results show that in educational 5G network, the clients’ behaviors are highly correlated with the campus activities." The quality of the manuscript will be enhanced by removing them. 

Other considerations: 

(3) L243: ’1+1+1+N’ system architecture accord with the OSI’s 7-layer network logic, which is convenient for unified industry understanding. -> This sentence is malformed and makes no sense in the context. Please update and elaborate.

(4) L246: in the following section 3.1.1-3.1.4 -> in the following sections 3.1.1-3.1.4

(5) L289: remove "etc"

(6) L347: remove "etc"

(7) L348: "The proposed architecture provides better support and availability for the above applications." -> How?

(8) L442: The 5G & WiFi -> The 5G and WiFi

(9) L595: remove "etc."

(10) L626: classroom/lab -> classroom or laboratory

(11) L734: remove "etc"

(12) L706: the app -> the application

(13) Table 1. Abbreviations used throughout this paper. -> Should be placed at the end of the paper.

(14) Table 2. Functionality Comparison between traditional and proposed architectures. -> To us is not sound to present a list of supported items that are all satisfied by architecture, and partially or not supported by another. Can we add additional criteria, to understand where the eMEC architecture is not good or not good enough? Perhaps there is another way to present this data more clearly. Either redrawing the table (we do not need all cells of the eMEC architecture with a "yes" value) or presenting this information within the text.

(15) Table 3: fibre -> fiber optic -> please revise this term across the manuscript.

(16) We consider the authors should review the manuscript to ensure no informal words and expressions are used across the manuscript.

Author Response

Thank your very much for your thorough review and constructive comments. We have carefully addressed the review comments and improved the quality of this paper. Please find below our response to review comments point-by-point.

Reviewer 3 Report

The authors addressed almost all my concerns in their revised paper, they made significant changes that improved the article. I still recommend to redraw figure 10 to be stylistically the same as the other ones.

I recommend to accept the paper in this present form.

Author Response

Thank you for your positive and constructive comments. We have revised Figure 10 to be stylistically the same as the other ones.

Round 3

Reviewer 1 Report

The authors provided all the required information. In this version, I recommend the paper publication. Regards.